# Assessment of Radiation Dose from the Consumption of Bottled Drinking Water in Japan

**DOI:** 10.3390/ijerph17144992

**Published:** 2020-07-11

**Authors:** Aoife Kinahan, Masahiro Hosoda, Kevin Kelleher, Takakiyo Tsujiguchi, Naofumi Akata, Shinji Tokonami, Lorraine Currivan, Luis León Vintró

**Affiliations:** 1Environmental Protection Agency, Clonskeagh Square, Clonskeagh, D14 H424 Dublin 14, Ireland; K.Kelleher@epa.ie (K.K.); L.Currivan@epa.ie (L.C.); 2Graduate School of Health Science, Hirosaki University, 66-1 Hon-cho, Hirosaki, Aomori 036-8564, Japan; m_hosoda@hirosaki-u.ac.jp (M.H.); r.tsuji@hirosaki-u.ac.jp (T.T.); 3Institute of Radiation Emergency Medicine, Hirosaki University, 66-1 Hon-cho, Hirosaki, Aomori 036-8564, Japan; akata@hirosaki-u.ac.jp (N.A.); tokonami@hirosaki-u.ac.jp (S.T.); 4School of Physics, University College Dublin, Dublin 4, Ireland; luis.leon@ucd.ie

**Keywords:** dose assessment, Japan, bottled water, guidance level, WHO, natural radionuclides, artificial radionuclides, effective dose, ingestion

## Abstract

Activity concentrations of ^234^U, ^235^U, ^238^U, ^226^Ra, ^228^Ra, ^222^Rn, ^210^Po, ^210^Pb, ^40^K, ^3^H, ^14^C, ^134^Cs and ^137^Cs were determined in 20 different Japanese bottled drinking water commercially available in Japan. The origins of the mineral water samples were geographically distributed across different regions of Japan. Activity concentrations above detection limits were measured for the radionuclides ^234^U, ^235^U, ^238^U, ^226^Ra, ^228^Ra and ^210^Po. An average total annual effective dose due to ingestion was estimated for adults, based on the average annual volume of bottled water consumed in Japan in 2019, reported to be 31.7 L/y per capita. The estimated dose was found to be below the recommended World Health Organisation (WHO) guidance level of 0.1 mSv/y for drinking water quality. The most significant contributor to the estimated dose was ^228^Ra.

## 1. Introduction

People are continuously exposed, both externally and internally, to ionising radiation from naturally occurring sources, such as cosmic rays and terrestrial radioactivity, as well as artificial sources, such as weapons fallout and authorized and accidental releases from the nuclear industry. The main pathways for internal exposure to radiation are inhalation and the ingestion of food and water. On average, people typically receive an estimated dose of 0.3 mSv/y due to the ingestion of naturally occurring radionuclides such as ^238^U, ^232^Th, their daughter products and ^40^K in their diet [1]. Drinking water contributes to approximately 0.01 mSv of this dose [2]. While this is a small fraction of the overall ingested dose, drinking water needs to be monitored, as there can be large variations in the radioactivity content in water as a result of the underlying geology of the water source. Higher radioactivity concentrations in drinking water can result in a higher contribution to the overall ingested dose. The World Health Organisation (WHO) has issued guidelines for drinking water quality. If the radioactivity dose from drinking water is below the WHO recommended guidance level of 0.1 mSv/y, water is deemed safe for human consumption from a radiological perspective [3].

The volume of bottled water being consumed has increased steadily worldwide for the past 14 years, and this trend is also reflected in the consumption of bottled water in Japan. According to the report by the Mineral Water Association of Japan, the annual intake of bottled water consumed in Japan in 2019 was reported to be 31.7 L/y per capita [4]. As shown in Figure 1, this is still significantly lower than bottled water consumption in many EU countries and the USA. However, a significant increase in the consumption of bottled water has occurred in Japan since 2011 [4], which has been attributed to consumer behavior after the Fukushima Dai-ichi Nuclear Power Plant (FDNPP) accident.

In this study, an estimation is made of the radioactivity dose ingested from the consumption of bottled water in Japan, based on the measurement of activity concentrations for a range of radiologically significant natural and artificial radionuclides in a selection of 20 different bottled waters collected throughout the country. The inclusion of artificial radionuclides such as ^3^H, ^14^C (also naturally occurring), ^134^Cs and ^137^Cs is considered particularly important from a radiation risk communication perspective, particularly after the FDNPP accident in 2011. Currently, 33 nuclear reactors remain operable in Japan, of which 9, located in 5 separate power plants, are in operation [5].

A review of the scientific literature shows that a number of previous studies were carried out to determine the radioactivity levels in bottled water consumed in Japan. For example, Shiraishi et al. investigated the effective dose due to ^232^Th and ^238^U in imported mineral waters [6], and Shozugawa analysed four Japanese bottled waters for ^134^Cs and ^137^Cs following the FDNPP accident in 2011 [7]. However, no single study has attempted to analyse Japanese bottled waters for a comprehensive suite of radionuclides, although similar studies have been conducted in other countries for estimating the activity concentration and ingestion doses arising from the consumption of natural radionuclides in bottled water [8,9,10,11,12,13,14,15,16,17,18,19,20,21,22].

The annual effective dose arising from the consumption of bottled water is estimated in this study, which represents the first comprehensive study of this kind to be undertaken in relation to Japanese bottled water. The results presented are based on the analysis of 20 Japanese bottled water samples for natural and artificial radionuclides.

## 2. Materials and Methods 

### 2.1. Sampling and Chemical Characteristics of Bottled Water

Drinking water samples from 20 different bottled water companies were purchased from Japanese supermarkets. The water sources were distributed across the country of Japan (Figure 2), which gives a wider geographical distribution to capture any potential effects of variability of the underlying geology of the regions. To increase representativeness, multiple bottles were purchased for each bottled water company and combined into a single sample. All water samples originated from natural mineral water sources.

To determine the chemical characteristics of the bottled water, pH and water hardness were evaluated using a pH meter (Eutech pH 700, Thermo Fisher Scientific, Waltham, MA, USA) and by determining the contents of magnesium (Mg) and calcium (Ca) by ICP–MS (ELAN 9000, Perkin Elmer, Germany), respectively. The water hardness (CaCO_3_ content) was calculated using Equation (1).
Total Hardness (mg/L as CaCO_3_) = 2.5Ca + 4.1Mg(1)

According to WHO guidelines [23], waters are classified on the basis of their hardness into soft (CaCO_3_ < 60), moderately hard (60 < CaCO_3_ < 120), hard (120 < CaCO_3_ < 180) and very hard (CaCO_3_ > 180).

### 2.2. Radionuclide Determination

All samples were analysed for radioactivity in the Environmental Protection Agency, Ireland’s radiation monitoring laboratory, which is accredited to ISO 17025 [24]. Table 1 gives a summary of the radionuclides analysed and the analytical techniques used for their determination. The methods and associated references are discussed in more detail below.

#### 2.2.1. Tritium and Carbon-14 Determination

For the determination of ^3^H and ^14^C, the spiked duplicate method was employed. Eight (8) mL of each water sample was taken for each radionuclide and mixed with 12 mL of Ultima Gold™ LLT (Perkin Elmer, Groningen, The Netherlands) liquid scintillation cocktail. A duplicate sample was made with 7 mL of water sample and 1 mL of a calibrated ^3^H or ^14^C standard and mixed with 12 mL of Ultima Gold™ LLT. These samples were analysed for 720 min using a Tri-Carb^®^ 3170TR/SL (Perkin Elmer, Waltham, MA, USA) low-level liquid scintillation counter. Analysis of the sample and the spiked duplicate sample was used to account for chemical and colour quenching in each sample. Optimised counting parameters were used with energy windows of 0.0–18.6 keV for ^3^H and 0.0–156.0 keV for ^14^C.

#### 2.2.2. Radon Determination

For the determination of ^222^Rn, 8 mL of sample was poured into a 20 mL scintillation vial containing Perkin Elmer, Ultima Gold™ F liquid scintillation cocktail. The vial was shaken for 60 s and counted on a Perkin Elmer Tri-Carb^®^ 3170TR/SL low-level liquid scintillation counter for 60 min, after 4 h of dark equilibration and to allow for the short-lived daughter products to reach secular equilibrium. These short-lived daughter products were measured to assess radon activity. The alpha energy window was optimised to 300–1000 keV. The principle of this method is based on the ISO standard, ISO 13164–Part 4 [25].

#### 2.2.3. Potassium and Caesium Determination

^40^K and ^134,137^Cs were analysed by direct counting on a gamma spectrometry system using in-house analytical methods accredited to the ISO 17025 standard [24]. The samples were measured with a high-purity germanium (HPGe) p-type co-axial detector (GC7520/S Mirion Technologies Inc., San Ramon, CA, USA). Radiocaesium concentrations in the water samples were determined by counting photons in the full-energy peak channels of 605 keV for ^134^Cs, 662 keV for ^137^Cs and 1461 keV for ^40^K. The measurement time was 86,400 s. The uncertainty for the activity concentration was evaluated taking into account the uncertainties of the counts for the sample and background. Coincidence summing, self-attenuation and decay corrections were applied using Canberra ApexGamma software (Mirion Technologies Inc., San Ramon, CA, USA) in conjunction with GESPECOR Monte-Carlo software (CID Media GmbH, Hasselroth, Germany).

The stable potassium content was also determined by ICP–MS (ELAN 9000, Perkin Elmer, Waltham, MA, USA). Fifty (50) mL of water sample was preserved in 0.5%v/v nitric acid. The principle of this method is based on the ISO standard, ISO 17294-2:2016 [26]. The isotopic abundance of ^40^K in natural potassium is 0.0117%, where ^40^K has a specific activity concentration of 3.1 × 10^−2^ Bq/kg. The amount of K-40 present in the sample was calculated by multiplying the amount of potassium measured (mg/L) by the activity concentration (Bq/kg) and isotopic abundance.

#### 2.2.4. Uranium Determination

The activity concentrations of ^234,235,238^U were determined by ICP–MS (ELAN 9000, Perkin Elmer, Waltham, MA, USA). Fifty (50) mL of water sample was preserved in 0.5% v/v nitric acid. The total uranium content was assessed. The principle of this method is based on the ISO standard, ISO 17294-2:2016 [26] Assuming that the uranium isotopes are present in their natural isotopic abundances in the environment (^234^U = 0.0055%, ^235^U = 0.72%, ^238^U > 99%), the activity concentration of ^234^U, ^235^U and ^238^U were determined [21]. 

#### 2.2.5. Radium Determination

For the determination of ^226^Ra and ^228^Ra, between 1.5 and 4 L was taken for the analysis. Each sample was acidified with nitric acid to a pH between 0 and 1. Thirty (30) mL of concentrated hydrochloric acid and 50 mg barium carrier solution were added to the sample in the beaker. The sample solution was boiled for 10 min. Thirty (30) mL of 9 M sulphuric acid was added and further boiled for 30 min to form a precipitate. The solution was left to cool overnight to allow the precipitation and settling of barium sulphate. The solution was then filtered through a pre-weighed filter paper (Whatman GF/C, 47 mm) (Whatman plc., Maidstone, United Kingdom). The filter was washed with 10 mL of ethanol and 10 mL of diethyl ether. The filter paper was allowed to dry overnight in a desiccator. The filter paper was prepared in a tightly sealed container, and the daughter products ^214^Pb, ^214^Bi (^226^Ra) and ^228^Ac (^228^Ra) were ingrown for more than 30 days to reach secular equilibrium with Ra. The measurements were made via HPGe gamma spectrometry. GESPECOR Monte-Carlo software (CID Media GmbH, Hasselroth, Germany) is typically used for correction for true coincidence summing and self-attenuation of the samples; however, these were not required in this instance [27].

#### 2.2.6. Polonium and Lead Determination

For the determination of ^210^Po and ^210^Pb, samples were prepared for spontaneous deposition on a silver (Ag) disc. An aliquot of 500 mL of water sample was placed in a jar and acidified with 2 M HCl (pH of 1–2) followed by the addition of 0.5 g of ascorbic acid and 0.25 Bq of ^209^Po tracer. A silver disc was fixed to a holder attached to the lid of the jar, which allowed the disc to be immersed in the solution. The solution was heated to 80 °C with stirring for 8 h. The disc was rinsed with water and methanol and dried at room temperature. The ^210^Po activities were determined by counting the disks for 345,600 s using PIPS detectors in the Canberra Alpha Analyst (Mirion Technologies Inc., San Ramon, California, USA) integrated spectrometry system in conjunction with the Apex Alpha spectrometry software (Mirion Technologies Inc., San Ramon, California, USA) [28]. The water samples were stored for 6 months, to allow the ingrowth of the ^210^Pb daughter product, ^210^Po. The spontaneous deposition process was repeated with a new aliquot of ^209^Po tracer. After analysing the measurements by alpha spectrometry, the ^210^Pb activity concentrations were determined from the polonium activities using Bateman’s equations [29].

### 2.3. Determination of the Annual Effective Ingestion Doses

The annual effective dose for each radionuclide was estimated using the following equation:*E_d_ = A_c_ × C_a_ × D_coeff_*(2)
where *E**_d_* is the annual effective dose due to ingestion (nSv); *A**_c_* is the radionuclide activity concentration measured in the water sample (Bq/L); *C**_a_* is the annual consumption of 31.7 L (per capita) reported by the Japanese Mineral Association [4]; and *D**_coeff_* is the dose coefficient provided by the ICRP (nSv/Bq) [30], whose values are given in Table 2. To get the total effective dose for a given sample, the effective doses for each radionuclide were added. The effective dose was estimated for each sample based on the activities of the radionuclides analysed. For measurements below the minimum detectable activity concentration (MDC), the MDC value was used for the dose estimation, which gave a maximum annual effective dose for each sample.

## 3. Results and Discussion

### 3.1. pH and Water Hardness

The pH of the 20 bottled water samples ranged from 6.2 to 8.3. The water hardness, which was equivalent to the CaCO_3_ content, ranged from below the detection limit (<4 mg/L) to 147 mg/L. Only one sample (No. 18) was classified as hard water. Samples 1, 3 and 17 were classified as moderately hard water, and the remaining Japanese bottled water samples were classified as soft water. No apparent differences were observed between soft and hard water samples.

### 3.2. Radionuclide Activity Concentrations

The activity concentrations measured in the bottled water samples, referenced to the date of receipt at the EPA laboratory, are given in Table 3. ^222^Rn was not expected to be present in bottled water, as during the bottling process radon gas is released from the product. The analysis was conducted to rule out any ingrowth from its parent radionuclide, ^226^Ra. Elevated levels of ^3^H and ^14^C were not expected either, as sources of these radionuclides discharged into the catchments of the bottled water sources are not significant or present in the environment at levels above the limits of detection for these radionuclides. This was confirmed by our measurements, which showed ^3^H and ^14^C activity concentrations below the detection limits in all cases. There were no detectable quantities of ^134^Cs and ^137^Cs in the water samples either. This indicates that the bottled water is not affected by fallout originating from the FDNPP accident. The MDC of ^222^Rn, ^3^H, ^14^C, ^134^Cs and ^137^Cs were determined as 60, 210, 160, 10 and 9 mBq/L, respectively. ^40^K determined by gamma analysis had an MDC of 160 mBq/. Therefore, the quoted ^40^K activity concentrations are based on the determination of stable K by ICP–MS and were calculated based on its natural abundance in the environment.

#### 3.2.1. Polonium and Lead activity Concentrations

In Table 4, the minimum, maximum and median activity concentrations of ^210^Po and ^210^Pb of the detected (above MDC) concentrations are shown. The maximum estimated annual effective dose, determined using Equation (2), is also included, which is based on the maximum measured activities for each nuclide. Seven bottled water samples showed ^210^Po concentrations above the MDC. For ^210^Pb, 19 samples were above the MDC.

#### 3.2.2. Radium Isotope Activity Concentrations

In Table 5, the minimum, maximum and median values of the radium isotopes of the detected concentrations are shown. The maximum estimated annual effective dose using Equation (2) is also included, which is based on the measured activities. Four and five bottled water samples showed concentrations of ^226^Ra and ^228^Ra above the MDC, respectively.

#### 3.2.3. Uranium Isotope Activity Concentrations

In Table 6, the minimum, maximum and median values of the uranium isotopes of the detected concentrations are shown. The maximum estimated annual effective dose, determined using Equation (2), is also included, which is based on the measured activities. Eighteen bottled water samples contained detectable amounts of total uranium, and two were below the limits of detection. The total uranium content of the bottled water was measured, and the activity concentrations were calculated based on their natural abundance in the environment.

#### 3.2.4. Comparison of Activity Concentrations in Bottled Mineral Waters for Sale in Other Countries

The activity concentrations found in the bottled water samples in this study were compared to those determined in previous studies in different countries. Most studies included the analysis of radium and uranium. In general, most studies reported radium as the largest contributor to the ingestion dose, apart from Rozmaric et al. [20] who reported uranium isotopes as the main contributor in Croatian mineral waters. The activity concentrations measured in this work for ^210^Po were 1.0–4.9 mBq/L. These values are comparable with results found in Croatia [20] but are lower than those reported in Italy [14,15] and Austria [10,19]. Activity concentrations for ^210^Pb were 0.53–19 mBq/L, whereas studies in Austria [10,19] showed a broader range for these values but reported a similar geometric mean for ^210^Pb activities. In contrast, values reported for Croatian waters [20] showed a lower range of activities. The activity concentrations for ^226^Ra and ^228^Ra ranged between 0.85 and 13 mBq/L and 16 and 49 mBq/L, respectively, which are comparable to the values found in Greece [12] and Pakistan [13]. Studies in Algeria [9], Croatia [20], Italy [14,15] and Poland [16] reported a much broader range of ^226^Ra and ^228^Ra activity concentrations in bottled water, but all geometric mean values are in agreement with the values reported in this study, with the exception of those for Algeria [9]. Algeria reported a mean value of 26 mBq/L. Studies in Austria [10,19] reported a broader range in radium activity concentrations and a higher mean value than this study. Studies in Malaysia [22] and Hungary [11] reported a very large range for ^226^Ra, with some values reported above 3000 mBq/L. For uranium isotopes, the ranges of activities reported in Japanese bottled water in this study were 0.24–24 mBq/L and 0.13–26 mBq/L for ^238^U and ^234^U, respectively. Studies in Poland [16] reported a lower range in uranium activity concentrations, whereas Croatia [20] and Tunisia [18] reported similar activities. Studies in Italy [14,15] and Austria reported a broader range in their results. In both Italian studies, the geometric mean values for ^234^U, ^235^U and ^238^U were much higher than the reported geometric mean values in this study [14,15]. 

It has been reported that the absorbed dose rates in air due to terrestrial radiation in western parts of Japan are higher than those in eastern parts of Japan [32]. This difference in dose rate distribution has been attributed to the presence in western Japan of areas with higher natural uranium and actinium series radionuclides, corresponding to granitic regions with enhanced natural radionuclide concentrations, relative to other areas characterised by andesite or andosol soils. To investigate a possible similar effect on the uranium content between waters sourced in western and eastern Japan, the median activity concentrations were statistically compared. The median values of ^234^U, ^235^U and ^238^U in western parts of Japan (Nos. 14, 15, 16, 17, 18, 19 and 20) were evaluated as 0.17 mBq/L, 0.063 mBq/L and 1.3 mBq/L, respectively. The corresponding values in eastern parts of Japan were evaluated as 0.39 mBq/L, 0.017 mBq/L and 0.36 mBq/L, respectively. A Mann–Whitney U Test (with a critical value of U at *p* < 0.05) showed that the values for each radionuclide from western and eastern Japan were not significantly different from each other. 

### 3.3. Dose Assessment of the Maximum Annual Effective Ingestion Doses from Bottled Water

According to the WHO, ^40^K levels do not need to be measured when assessing the dose from drinking water. The potassium level in a healthy individual is kept constant by a range of physiological processes in order to regulate the functions of the body [2]. Therefore, the levels of ^40^K were not included in the calculation of the annual effective ingestion dose.

Figure 3 presents an overview of the estimated maximum annual effective dose for each sample dose received from the radionuclides analysed in the bottled water samples. The highest dose value calculated was 10% of the guidance level recommended by the WHO. The average annual effective dose estimated for all samples was 5.6 μSv. There was little correlation between doses and locations across Japan. There were two bottled water products from the same location, i.e., 9 and 10 and 7 and 8, and these reported similar dose values. In contrast, samples 12 and 13 were not similar, though they were from the same location.

## 4. Conclusions

The activity concentrations of natural and artificial radionuclides in a wide range of bottled water produced in Japan were determined. The results showed that the highest activities were due to naturally occurring radionuclides, i.e., Ra and Pb isotopes. The results are comparable to results from other studies around the world. The total annual effective ingestion dose was assessed from the activity concentrations measured in this study and the average rate of consumption of bottled mineral water in Japan per capita. In all cases, the estimated doses were at least an order of magnitude below the WHO recommended guidance level of 0.1 mSv for the consumption of drinking water.

## Figures and Tables

**Figure 1 ijerph-17-04992-f001:**
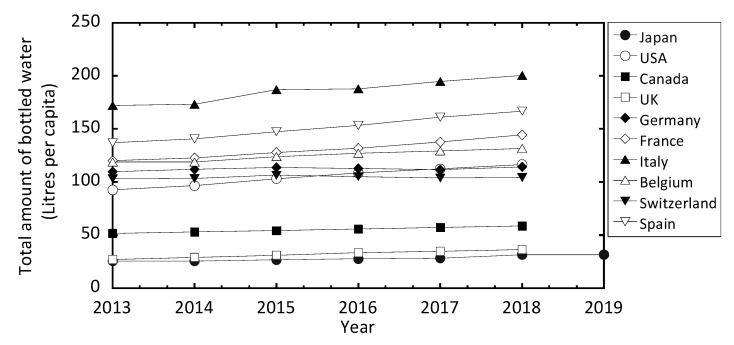
Trend graph for the consumption of bottled water in 10 different countries, including Japan [4].

**Figure 2 ijerph-17-04992-f002:**
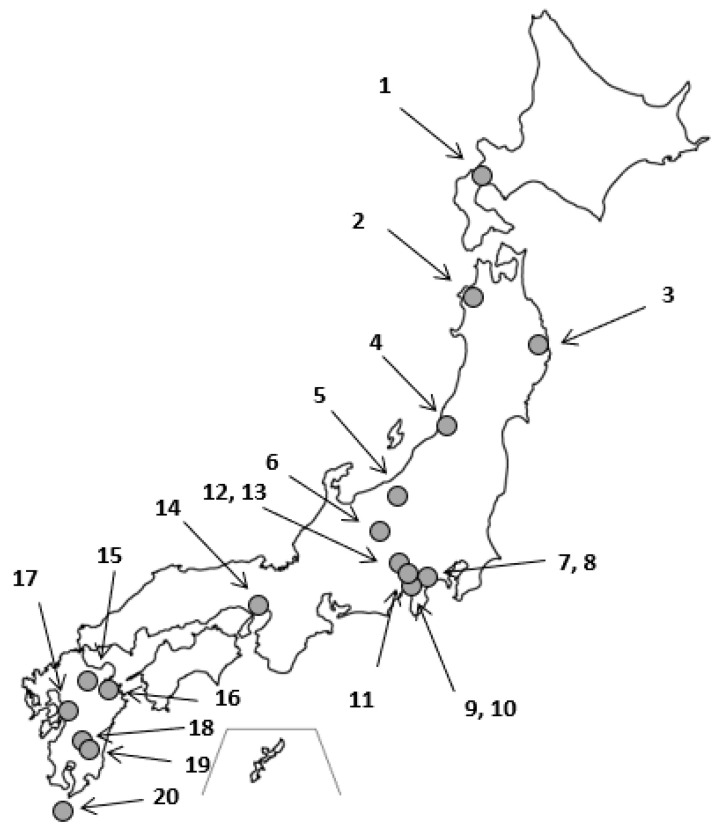
Locations of 20 bottle water samples.

**Figure 3 ijerph-17-04992-f003:**
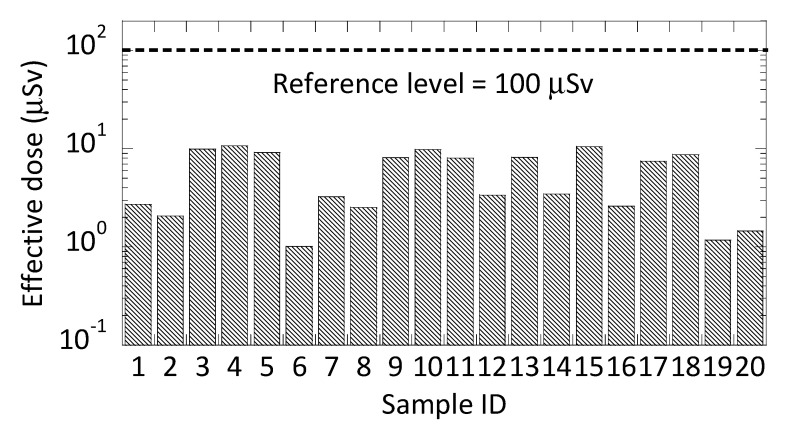
The maximum annual effective dose in each bottled water sample compared to the guidance level.

**Table 1 ijerph-17-04992-t001:** Summary of the methods used for the analysis. HPGe, high-purity germanium.

Radionuclide	Preparation	Analytical Techniques
^3^H, ^14^C	8:12 mL water:/Ultima Gold ™ LLT 7:1:12 mL water/spike/Ultima Gold ™ LLT	Liquid scintillation counting
^222^Rn	8:12 mL water/organic scintillation cocktail	Liquid scintillation counting
^40^K, ^134^Cs, ^137^Cs	500 mL Marinelli	Direct counting on HPGe
^40^K	Acidified	ICP–MS (evaluated from stable potassium to ^40^K)
^210^Po	500 mL water, acidified, spontaneous deposition	Alpha spectrometry
^210^Pb	500 mL water, acidified, spontaneous deposition (after 6 months)	Alpha spectrometry
^226^Ra, ^228^Ra	1 to 4 L water, barium co-precipitation method, stored for one month prior to measurement	Gamma spectrometry (HPGe)
^234^U, ^235^U, ^238^U	Acidified	ICP–MS (evaluated from total uranium)

**Table 2 ijerph-17-04992-t002:** Dose coefficient values used for this study (nSv/Bq).

Isotope	Dose Coefficient (nSv/Bq)
^222^Rn [31] ^1^	3.5 × 10^0^
^3^H	1.8 × 10^−2^
^14^C	5.8 × 10^−1^
^134^Cs	1.9 × 10^1^
^137^Cs	1.3 × 10^1^
^210^Po	1.2× 10^3^
^210^Pb	6.9 × 10^2^
^226^Ra	2.8 × 10^2^
^228^Ra	6.9 × 10^2^
^234^U	4.9 × 10^1^
^235^U	4.7 × 10^1^
^238^U	4.5 × 10^1^

^1^ Specific to ^222^Rn in drinking water, the value is the sum of weighted equivalent doses.

**Table 3 ijerph-17-04992-t003:** Radionuclide activity concentration and standard uncertainty (*k* = 1) values in bottled water (mBq/L).

Sample	^210^Po	^210^Pb	^226^Ra	^228^Ra	^234^U	^235^U	^238^U	^40^K
1	<9.4 × 10^−1^	(5.7 ± 0.5) × 10^0^	<1.9 × 10^2^	(2.6 ± 1.5) × 10^1^	(2.6 ± 0.5) × 10^1^	(1.2 ± 0.2) × 10^0^	(2.4 ± 0.5) × 10^1^	(1.3 ± 0.3) × 10^−1^
2	<7.4 × 10^−1^	(1.8 ± 0.2) × 10^0^	<1.1 × 10^2^	<3.7 × 10^1^	<1.3 × 10^−1^	<5.8 × 10^−3^	<1.2 × 10^−1^	<7.7 × 10^−3^
3	<1.3 × 10^0^	(2.0 ± 0.2) × 10^0^	<1.8 × 10^2^	<3.6 × 10^2^	(2.4 ± 0.5) × 10^0^	(1.0 ± 0.2) × 10^−1^	(2.2 ± 0.5) × 10^0^	(1.5 ± 0.3) × 10^−2^
4	(2.6 ± 0.4) × 10^0^	(7.4 ± 0.7) × 10^0^	<2.0 × 10^2^	<3.9 × 10^2^	(2.6 ± 0.5) × 10^−1^	(1.2 ± 0.2) × 10^−2^	(2.4 ± 0.5) × 10^−1^	(1.9 ± 0.4) × 10^−2^
5	<1.2 × 10^0^	< 1.7 × 10^0^	<1.7 × 10^2^	<3.4 × 10^2^	(1.3 ± 0.3) × 10^−1^	(5.8 ± 0.1) × 10^−3^	(1.2 ± 0.3) × 10^−1^	(3.8 ± 0.8) × 10^−2^
6	<1.3 × 10^0^	(5.3 ± 0.5) × 10^0^	(8.5 ± 3.9) × 10^0^	(2.5 ± 1.1) × 10^1^	(7.9 ± 0.2) × 10^−1^	(3.5 ± 0.7) × 10^−2^	(7.3 ± 0.2) × 10^−1^	(6.2 ± 1.3) ± × 10^−2^
7	<8.9 × 10^−1^	(3.8 ± 0.3) × 10^0^	<2.3 × 10^2^	<4.2 × 10^1^	(5.2 ± 1.1) × 10^−1^	(2.3 ± 0.5) × 10^−2^	(4.8 ± 1.0) × 10^−1^	(3.5 ± 0.7) × 10^−2^
8	<1.2 × 10^0^	(5.3 ± 0.5) × 10^−1^	<1.7 × 10^2^	<3.4 × 10^1^	(2.6 ± 0.5) × 10^−1^	(1.2 ± 0.2) × 10^−2^	(2.4 ± 0.5) × 10^−1^	(4.9 ± 1.0) × 10^−2^
9	<1.1 × 10^0^	(3.7 ± 0.3) × 10^0^	<6.0 × 10^2^	<1.1 × 10^2^	(1.3 ± 0.3) × 10^−1^	(5.8 ± 1.2) × 10^−3^	(1.2 ± 0.3) × 10^−1^	(2.5 ± 0.5) × 10^−2^
10	<1.1 × 10^0^	(8.1 ± 0.7) × 10^−1^	<1.1 × 10^2^	<3.9 × 10^2^	(1.3 ± 0.3) × 10^−1^	(5.8 ± 1.2) × 10^−3^	(1.2 ± 0.3) × 10^−1^	(2.5 ± 0.5) × 10^−2^
11	<5.7 × 10^−1^	(2.0 ± 0.2) × 10^0^	<1.5 × 10^2^	<2.9 × 10^2^	(3.9 ± 0.8) × 10^−1^	(1.7 ± 0.4) × 10^−2^	(3.6 ± 0.8) × 10^−1^	(2.4 ± 0.5) × 10^−2^
12	(1.0 ± 0.3) × 10^0^	(4.4 ± 0.4) × 10^0^	<2.4 × 10^2^	<4.3 × 10^1^	(5.2 ± 0.1) × 10^−1^	(2.3 ± 0.5) × 10^−2^	(4.8 ± 1.0) × 10^−1^	(3.2 ± 0.7) × 10^−2^
13	(4.9 ± 0.4) × 10^0^	(1.9 ± 0.2) × 10^1^	(1.2 ± 0.5) × 10^1^	<3.3 × 10^2^	(1.6 ± 0.3) × 10^0^	(6.9 ± 0.1) × 10^−2^	(1.5 ± 0.3) × 10^0^	(5.1 ± 1.1) × 10^−2^
14	(1.5 ± 0.6) × 10^0^	(4.3 ± 0.4) × 10^0^	<2.3 × 10^2^	(4.9 ± 1.8) × 10^1^	<1.3 × 10^−1^	<5.8 × 10^−3^	<1.2 × 10^−1^	(2.9 ± 0.6) × 10^−2^
15	<1.4 × 10^0^	(3.0 ± 0.3) × 10^0^	<1.9 × 10^2^	<3.9 × 10^2^	(3.0 ± 0.6) × 10^0^	(1.3 ± 0.3) × 10^−1^	(2.8 ± 0.6) × 10^0^	(2.5 ± 0.5) × 10^−1^
16	<9.7 × 10^−1^	(4.5 ± 0.4) × 10^0^	<1.9 × 10^2^	(1.9 ± 1.0) × 10^1^	(1.4 ± 0.3) × 10^−1^	(6.3 ± 1.3) × 10^−2^	(1.3 ± 0.3) × 10^0^	(8.4 ± 1.7) × 10^−2^
17	<8.3 × 10^−1^	(2.4 ± 0.2) × 10^0^	(1.3 ± 0.6) × 10^1^	<3.2 × 10^2^	(2.6 ± 0.5) × 10^−1^	(1.2 ± 0.2) × 10^−2^	(2.4 ± 0.5) × 10^−1^	(1.4 ± 0.3) × 10^−1^
18	<1.4 × 10^0^	(2.6 ± 0.2) ×10^0^	<1.6 × 10^2^	<3.2 × 10^2^	(2.8 ± 0.6) × 10^0^	(1.2 ± 0.2) × 10^−1^	(2.5 ± 0.5) × 10^0^	(2.4 ± 0.5) × 10^−1^
19	(1.9 ± 1.5) × 10^0^	(6.9 ± 0.6) × 10^0^	(1.2 ± 0.4) × 10^1^	<2.5 × 10^1^	(1.7 ± 0.4) × 10^−1^	(7.5 ± 1.5) × 10^−2^	(1.6 ± 0.3) × 10^0^	(7.2 ± 1.5) × 10^−2^
20	(1.7 ± 1.5) × 10^0^	(5.9 ± 0.5) × 10^0^	<7.1 × 10^2^	(1.6 ± 0.8) × 10^1^	(1.3 ± 0.3) × 10^−1^	(5.8 ± 1.2) × 10^−3^	(1.2 ± 0.3) × 10^−1^	(2.5 ± 0.5) × 10^−2^

**Table 4 ijerph-17-04992-t004:** Activity concentrations for polonium and lead measured in bottled water (mBq/L).

Isotope	Min	Max	Median	Maximum Annual Effective Dose ^1^ (nSv)
^210^Po ^2^	1.0 ± 0.26	4.9 ± 0.39	1.7 ± 0.34	186
^210^Pb ^3^	0.53 ± 0.050	19 ± 1.8	3.8 ± 0.34	421

^1^ Based on measured activities; ^2^ minimum detectable activity concentrations (MDC) = 0.06 mBq/L; ^3^ MDC = 0.12 mBq/L.

**Table 5 ijerph-17-04992-t005:** Activity concentrations for radium measured in bottled water (mBq/L).

Isotope	Min	Max	Median	Maximum Annual Effective Dose ^1^ (nSv)
^226^Ra ^2^	0.85 ± 0.39	13 ± 5.5	12 ± 4.3	100
^228^Ra ^3^	16 ± 8.2	49 ± 18	25 ± 11	589

^1^ Based on measured activities; ^2^ MDC = 112 mBq/L; ^3^ MDC = 25 mBq/L.

**Table 6 ijerph-17-04992-t006:** Activity concentrations for uranium analysed in bottled water (mBq/L).

Isotope	Min	Max	Median	Maximum Annual Effective Dose ^1^ (nSv)
^234^U ^2^	0.13 ± 0.030	26 ± 5.4	0.52 ± 0.11	3.3
^235^U ^3^	0.005 ± 0.002	1.2 ± 0.24	0.023 ± 0.005	0.14
^238^U ^4^	0.24 ± 0.050	24 ± 5.0	0.48 ± 0.099	2.8

^1^ Based on measured activities; ^2^ MDC = 0.13 mBq/L; ^3^ MDC = 0.005 mBq/L; ^4^ MDC = 0.12 mBq/L.

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
