# Peer review of "Assessment of Radiation Dose from the Consumption of Bottled Drinking Water in Japan"

_ijerph, 2020, doi:10.3390/ijerph17144992_

Round 1

Reviewer 1 Report

Overall comments:

  1. All References are present, appropriate and properly cited
  2. Equation 1 and 2 seem correct
  3. Proper use of significant figures
  4. Title and Abstract are appropriate
  5. Materials and Methods:
    1. The authors mention that drinking water samples from 20 different bottled water companies were purchased from Japanese supermarkets. Was only one sample of each purchased? If not, how many samples (bottles) were purchased?  Were the lot numbers the same, or different? If the 20 drinking water samples included multiple samples from the same company or brand (but from different sources), this should be stated. Overall the methodology is very sound and  well done but if only one “sample” was measured, this drastically reduces the power of the study given potential variations in water sources. 
    2. Methods for analysis and duplicate sample methodology (including spikes) are correct and state-of-the-art.
  6. Sentences should not begin with a numeric (for example, line 131, 133, “30 ml…”). Recommend changing (for example, line 131, 133, “Thirty (30) ml)…”
  7. Line 159: “concent” should be “content”
  8. The Results and Discussion are well explained and presented clearly
  9. Tables and Figures are appropriate
  10. Conclusion is brief but adequately summarizes the study.

This is an extremely well written article in which the authors clearly present relevant background, methodology, results, and conclusions related to the study.  The authors more than adequately cite the importance of their work, include the relevant literature related to the work, and show how work is different from previous studies. The methodology and analyses used are state-of-the-art and correctly applied. 

Reviewer 2 Report

The paper presents an assessment of the annual total doses of a set of radiologically significant radionuclides concentrations in bottled drinking water in Japan. Samples from different regions were analysed, and all estimated doses were found to be below the recommended WHO's guidance level.

In general, the work is well written and easy to follow. From a methodological point of view, the study makes no significant contribution, as it applies standard techniques to measure radionuclide concentrations and evaluate radiation doses. Nonetheless, in comparison with similar works, it comprises a broader geographical coverage and a more comprehensive set of radionuclides, providing a relevant contribution for future researchers and decision makers in public health.

Major comment:
It is not clear if the reported average volume of bottled water consumed per capita in Japan (31.7 L/y) takes into account the overall population or only the subpopulation of bottled water consumers. If this statistics takes into account the total population, it may be misleading and may induce a severe dose underestimation for the specific subpopulation of interest, compromising the main conclusions. Authors should clarify this point.

Minor comments:
- The description of the method for ingestion dose assessment (lines 238-245) should be moved to Section 2 (Material and Methods) in order to make the interpretation of Tables 3-5 easier.
- The dose coefficient (Dcoeff) values used in the study could be displayed in a table, to allow the reader to reproduce the computations.

Reviewer 3 Report

The manuscript entitled “Assessment of Radiation Dose from the Consumption of Bottled Drinking Water in Japan” determined activity concentrations and an average total annual effective dose due to ingestion of 234U, 235U, 238U, 226Ra, 228Ra, 222Rn, 210Po, 210Pb, 40K, 3H, 14C, 134Cs and 137Cs in 20 different Japanese bottled drinking water commercially available in geographically distributed across the different regions of Japan. They clearly concluded that the estimated dose was found to be below the recommended World Health Organisation (WHO)’s guidance level of 0.1 mSv/y, according to the WHO’s guidelines for drinking water quality. Their approach and findings are very interesting and valid to IJERPH readers. I have only a few suggestions to improve the manuscript with minor revision.

1) The novelty, research purpose and results of this study are not clearly shown in the introduction section. Therefore, the introduction section should be improved more simply and clearly.

2) Why did the authors determine water hardness for this study? Are there any insights into the activity concentrations? If so, the authors should mention in introduction about it and provide the data (not only classification). In addition, the unit of water hardness should be clearly replaced (in equation) with “Total hardness (mg/L as CaCO3)”.

3) I personally feel that water samples should have some Sr2+ ions which significantly influence total hardness. I recommend that the authors should check the concentrations of Sr2+ ions.

4) Conclusion like “The consumption of bottled water would have to increase ten-fold to reach these effective dose limit” should be provided more clearly.
